# The Discrete Dipole Approximation: A Review

## Patrick Christian Chaumet

Institut Fresnel, Aix Marseille Univ, CNRS, Centrale Marseille, CEDEX 20, 13397 Marseille, France; patrick.chaumet@fresnel.fr

**Abstract:** There are many methods for rigorously calculating electromagnetic diffraction by objects of arbitrary shape and permittivity. In this article, we will detail the discrete dipole approximation (DDA) which belongs to the class of volume integral methods. Starting from Maxwell's equations, we will first present the principle of DDA as well as its theoretical and numerical aspects. Then, we will discuss the many developments that this method has undergone over time and the numerous applications that have been developed to transform DDA in a very versatile method. We conclude with a discussion of the strengths and weaknesses of the DDA and a description of the freely available DDA-based electromagnetic diffraction codes.

**Keywords:** electromagnetic simulation; DDA; numerical method; electromagnetic scattering

## 1. Introduction

The interaction of an electromagnetic wave with matter is a broad field of application in physics. Although this interaction is entirely governed by Maxwell's equations, which have been known for 150 years, it remains an active subject of study, particularly the study of the interaction between light and particles ranging subwavelength in size from several hundred wavelengths.

Indeed, the rigorous calculations of light diffraction started earlier with spherical particles, where the theory developed by Mie in 1908 allowed solving the problem analytically [1,2]. Unfortunately, when the particles have more complicated shapes, there are no analytical solutions. It is then necessary to solve electromagnetic scattering rigorously with numerical methods.

In recent decades, many numerical methods have been developed. Among the best known are finite difference time domain (FDTD), finite element, volume integral methods, and surface integral methods. The interested readers can refer to [3,4] which provide a concise overview on some of the most widely used modern techniques for solving the electromagnetic scattering problem for particles of arbitrary shape.

In this article, we will focus on the volume integral method, and more specifically, the discrete dipole approximation (DDA) also called the coupled dipole method (CDM) or the coupled dipole approximation (CDA). This method was introduced in 1973 by Purcell and Pennypacker to study the scattering and absorption of light by non-spherical dielectric grains with dimensions comparable to the wavelength of illumination [5]. Since this seminal paper, DDA has been applied in a wide variety of fields such as astrophysics [6], near-field optics [7], biology [8,9], microscopy [10], optical diffraction tomography [11,12], holographic microscopy [13], optical force and tweezers [14–16]. The application of DDA in very different fields have led physicists to improve the DDA with new particularities and possibilities. For a overview of the DDA principle, the reader can refer to [17–19].

In this article, we will first explain the principle of DDA from Maxwell equations, and then detail the various advances made with the DDA over the last 20 years. We will end this article by detailing the strengths and weaknesses of the DDA and list the different electromagnetic diffraction codes based on DDA.

## 2. The Discrete Dipole Approximation

*2.1. Principle of the Discrete Dipole Approximation*

From Maxwell equations, the total electric field at $\mathbf{r} \in \mathbb{R}^3$ satisfies [2],

$$\boldsymbol{\nabla} \times [\boldsymbol{\nabla} \times \mathbf{E}(\mathbf{r})] - \varepsilon_{\text{ref}}(\mathbf{r}) k_0^2 \mathbf{E}(\mathbf{r}) = \mathbf{S} + 4\pi k_0^2 \overleftrightarrow{\chi}(\mathbf{r}) \mathbf{E}(\mathbf{r}), \tag{1}$$

where $k_0$ is the wavenumber in vacuum. Note that all the equations in this article are in Gaussian unit. $\mathbf{S}$ denotes the sources, which satisfies the homogeneous equation

$$\boldsymbol{\nabla} \times [\boldsymbol{\nabla} \times \mathbf{E}_{\text{ref}}(\mathbf{r})] - \varepsilon_{\text{ref}}(\mathbf{r}) k_0^2 \mathbf{E}_{\text{ref}}(\mathbf{r}) = \mathbf{S}, \tag{2}$$

with $\mathbf{E}_{\text{ref}}(\mathbf{r})$ the incident wave impinging on the object under study. $\overleftrightarrow{\chi}(\mathbf{r})$ is the electrical susceptibility of the object, a rank 2 tensor if it is anisotropic, and a scalar for an isotropic object. The total field $\mathbf{E}(\mathbf{r})$ is the sum of the reference and diffracted field, i.e., $\mathbf{E}_{\text{d}}(\mathbf{r}) = \mathbf{E}(\mathbf{r}) - \mathbf{E}_{\text{ref}}(\mathbf{r})$, the diffracted field satisfying the outgoing wave conditions. To calculate the total field, we introduce the Green tensor $\overleftrightarrow{\mathbf{G}}$, solution of

$$\boldsymbol{\nabla} \times \left[ \boldsymbol{\nabla} \times \overleftrightarrow{\mathbf{G}}(\mathbf{r}, \mathbf{r}') \right] - \varepsilon_{\text{ref}}(\mathbf{r}) k_0^2 \overleftrightarrow{\mathbf{G}}(\mathbf{r}, \mathbf{r}') = 4\pi k_0^2 \overleftrightarrow{\mathbf{I}} \delta(\mathbf{r} - \mathbf{r}'), \tag{3}$$

that satisfies outgoing boundary conditions. $\overleftrightarrow{\mathbf{I}}$ denotes the unit tensor of size $3 \times 3$ and $\varepsilon_{\text{ref}}$ the relative permittivity in the absence of the object. $\varepsilon_{\text{ref}}$ is a constant if the object is in a homogeneous space or depends on $z$ if the object is in the presence of a multilayer. The expression of the Green tensor in vacuum is [2]:

$$\overleftrightarrow{\mathbf{G}}(\mathbf{r}, \mathbf{r}') = e^{ik_0 R} \left[ \left( 3\hat{\mathbf{R}} \otimes \hat{\mathbf{R}} - \overleftrightarrow{\mathbf{I}} \right) \left( \frac{1}{R^3} - \frac{ik_0}{R^2} \right) + \left( \overleftrightarrow{\mathbf{I}} - \hat{\mathbf{R}} \otimes \hat{\mathbf{R}} \right) \frac{k_0^2}{R} \right], \tag{4}$$

where $\otimes$ denotes dyadic product. We have $\mathbf{R} = \mathbf{r} - \mathbf{r}'$, $R = |\mathbf{R}|$ and $\hat{\mathbf{R}} = \mathbf{R}/R$ and $\mathbf{r} \neq \mathbf{r}'$. In using the properties of the Green's function, the total field is then the solution of the following Lippmann–Schwinger equation [20]:

$$\mathbf{E}(\mathbf{r}) = \mathbf{E}_{\text{ref}}(\mathbf{r}) + \int_{\Omega} \overleftrightarrow{\mathbf{G}}(\mathbf{r}, \mathbf{r}') \overleftrightarrow{\chi}(\mathbf{r}') \mathbf{E}(\mathbf{r}') d\mathbf{r}', \tag{5}$$

where the integration is performed over the support $\Omega$ of the object under study (see Figure 1a). It means that $\overleftrightarrow{\chi}(\mathbf{r}) = 0$ for $\mathbf{r} \notin \Omega$ and $\overleftrightarrow{\chi}(\mathbf{r}) = \frac{\overleftrightarrow{\varepsilon}(\mathbf{r}) - \overleftrightarrow{\mathbf{I}}}{4\pi}$ for $\mathbf{r} \in \Omega$ where $\overleftrightarrow{\varepsilon}(\mathbf{r})$ is the relative permittivity of the object. To solve numerically the Lippmann–Schwinger equation, we discretize the object into a set of $N$ subunits arranged on an arbitrary orthogonal lattice (see Figure 1b). Hence, Equation (5) becomes:

$$\mathbf{E}(\mathbf{r}) = \mathbf{E}_{\text{ref}}(\mathbf{r}) + \sum_{j=1}^{N} \int_{V_j} \overleftrightarrow{\mathbf{G}}(\mathbf{r}, \mathbf{r}') \overleftrightarrow{\chi}(\mathbf{r}') \mathbf{E}(\mathbf{r}') d\mathbf{r}', \tag{6}$$

where $V_j$ is the volume of the subunit $j$.

To solve Equation (6) numerically, we need to make some approximations. The first is to assume that the electromagnetic field can be considered as uniform over a subunit, which is a good approximation if the subunit is smaller than the wavelength of the field inside the object. In that case, we can write:

$$\mathbf{E}(\mathbf{r}_i) = \mathbf{E}_{\text{ref}}(\mathbf{r}) + \sum_{j=1}^{N} \left( \int_{V_j} \overleftrightarrow{\mathbf{G}}(\mathbf{r}_i, \mathbf{r}') d\mathbf{r}' \right) \overleftrightarrow{\chi}(\mathbf{r}_j) \mathbf{E}(\mathbf{r}_j), \tag{7}$$

for $i = 1, \cdots, N$. The second approximation consists, for $i \neq j$, in considering the Green function as constant on a subunit, then we get:

$$\mathbf{E}(\mathbf{r}_i) = \mathbf{E}_{\text{ref}}(\mathbf{r}) + \sum_{j=1, j\neq i}^{N} \overleftrightarrow{\mathbf{G}}(\mathbf{r}_i, \mathbf{r}_j) V_j \overleftrightarrow{\chi}(\mathbf{r}_j) \mathbf{E}(\mathbf{r}_j) + \left( \int_{V_i} \overleftrightarrow{\mathbf{G}}(\mathbf{r}_i, \mathbf{r}') d\mathbf{r}' \right) \overleftrightarrow{\chi}(\mathbf{r}_i) \mathbf{E}(\mathbf{r}_i). \quad (8)$$

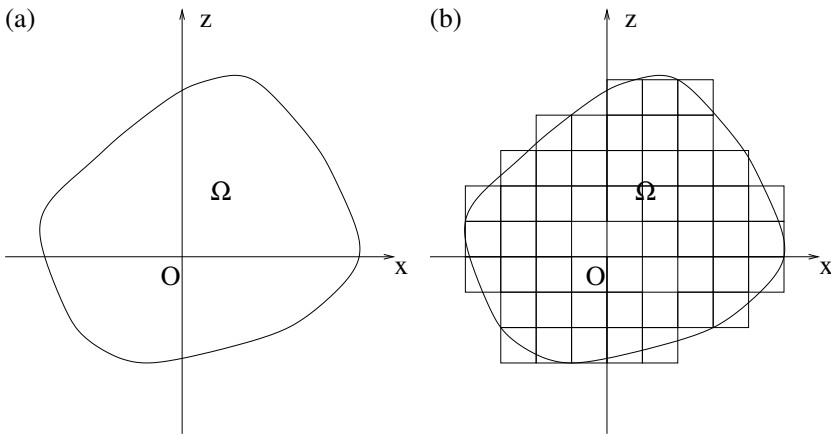

**Figure 1.** (**a**) Object with a support $\Omega$ in homogeneous background. (**b**) Discretization of the object with a cubic mesh for the DDA. The smaller the discretization mesh, the more the discretized object will have the shape of the studied object.

Note that in the case where the mesh is very small compared to the wavelength of illumination, it is then preferable to integrate numerically the Green's function for nearby subunits, i.e., for $|\mathbf{r}_i - \mathbf{r}_j| < \lambda/2$, to increase the accuracy of the DDA [21]. One of the key points of the DDA is to perform the integration of the diagonal term, i.e., $\int_{V_i} \overleftrightarrow{\mathbf{G}}(\mathbf{r}_i, \mathbf{r}') d\mathbf{r}'$. Assuming the subunit to be infinitesimally small and cubic, the integration of the diagonal term is written as [22]:

$$\lim_{V_i \to 0} \int_{V_i} \overleftrightarrow{\mathbf{G}}(\mathbf{r}_i, \mathbf{r}') d\mathbf{r}' = -\frac{4\pi}{3} \overleftrightarrow{\mathbf{I}}. \quad (9)$$

Note that if the subunit has a cuboid shape then the expression of the diagonal components of the tensor are different [22,23]. Hence, the final equation to solve for the DDA can be expressed as

$$\mathbf{E}_l(\mathbf{r}_i) = \mathbf{E}_{\text{ref}}(\mathbf{r}) + \sum_{j=1, j\neq i}^{N} \overleftrightarrow{\mathbf{G}}(\mathbf{r}_i, \mathbf{r}_j) \overleftrightarrow{\alpha_0}(\mathbf{r}_j) \mathbf{E}_l(\mathbf{r}_j), \quad (10)$$

$$\text{with} \quad \mathbf{E}_l(\mathbf{r}_i) = \mathbf{E}(\mathbf{r}_i) \frac{\overleftrightarrow{\varepsilon}(\mathbf{r}_i) + 2\overleftrightarrow{\mathbf{I}}}{3}, \quad (11)$$

$$\overleftrightarrow{\alpha_0}(\mathbf{r}_j) = \frac{3d^3}{4\pi} \left( \overleftrightarrow{\varepsilon}(\mathbf{r}_j) - \overleftrightarrow{\mathbf{I}} \right) \left( \overleftrightarrow{\varepsilon}(\mathbf{r}_j) + 2\overleftrightarrow{\mathbf{I}} \right)^{-1}, \quad (12)$$

where $\overleftrightarrow{\alpha_0}(\mathbf{r}_j)$ is the Clausius–Mossotti relationship for the subunit $j$, $d$ the lattice spacing of the cubic mesh, and $\mathbf{E}_l(\mathbf{r}_j)$ the local field for the subunit $j$, i.e., the field at the position $j$ due to the reference field plus the field radiated by the other subunits in the absence of the subunit $j$. We can note that $\mathbf{p}(\mathbf{r}_j) = \overleftrightarrow{\alpha_0}(\mathbf{r}_j) \mathbf{E}_l(\mathbf{r}_j)$ represents the dipole induced at the position $j$. Equations (10) and (12) correspond exactly to the DDA as presented in 1973 by Purcell and Pennypacker [5]. They have the disadvantage of not having conservation of energy because of the approximation made on the diagonal term of having an infinitely small size. If we take into account the finite size of the subunit $i$, and assuming that the

cube is equivalent to a sphere of the same volume, then the DDA equations can be written as [6,24]:

$$\mathbf{E}_l(\mathbf{r}_i) = \mathbf{E}_{\text{ref}}(\mathbf{r}) + \sum_{j=1,j\neq i}^{N} \overleftrightarrow{\mathbf{G}}(\mathbf{r}_i,\mathbf{r}_j)\overleftrightarrow{\alpha}(\mathbf{r}_j)\mathbf{E}_l(\mathbf{r}_j), \tag{13}$$

$$\text{with } \overleftrightarrow{\alpha}(\mathbf{r}_j) = \overleftrightarrow{\alpha_0}(\mathbf{r}_j)\left[\overleftrightarrow{\mathbf{I}} - \frac{2}{3}ik_0^3\overleftrightarrow{\alpha_0}(\mathbf{r}_j)\right]^{-1}. \tag{14}$$

The introduction of this imaginary part in the polarizability is commonly referred as the radiative reaction term and is essential to respect the optical theorem. It is of course possible to calculate the diagonal term rigorously (see [21,24]) or to use more accurate approximations [24]. For more details on how to compute the diagonal term (see Appendix A). Many studies have been made to improve the accuracy of the DDA by changing the form of the polarizability [25–30]. We can also increase the accuracy of the DDA for high refractive indices by changing the Green's function to the filtered coupled dipoles [7,18]. A study of the convergence has been conducted in [31,32] and numerous comparison with other numerical method have been conducted [9,33–36].

Then the system of linear equations represented by Equation (14), can be written symbolically as:

$$\mathbf{E}_l = \mathbf{E}_{\text{ref}} + \mathbf{A}\mathbf{D}_\alpha\mathbf{E}_l, \tag{15}$$

where $\mathbf{A}$ is a square matrix of size $(3N \times 3N)$ and contains all the dyadic tensors, $\mathbf{E}_l$ and $\mathbf{E}_{\text{ref}}$ are $3N$ vectors that contain the local and reference field, respectively, at the position of each subunit for the three components, and $\mathbf{D}_\alpha$ is a diagonal matrix of size $(3N \times 3N)$ which contain the polarizabilities if the permittivity is scalar and a block diagonal matrix such that the main-diagonal blocks are $3 \times 3$ matrices if permittivity is anisotropic. Once the Lippmann–Schwinger equation is solved, Equation (15), we can compute quickly the near field around the particle (see [37]), and we can calculate the diffracted far-field by the object at the position $\mathbf{r}$ with:

$$\mathbf{E}_d(\mathbf{r}) = \sum_{i=1}^{N} \overleftrightarrow{\mathbf{G}}_{\text{ff}}(\mathbf{r},\mathbf{r}_i)\overleftrightarrow{\alpha}(\mathbf{r}_i)\mathbf{E}_l(\mathbf{r}_i), \tag{16}$$

$$\text{with } \overleftrightarrow{\mathbf{G}}_{\text{ff}}(\mathbf{r},\mathbf{r}_i) = \frac{k_0^2}{r}\left(\overleftrightarrow{\mathbf{I}} - \hat{\mathbf{r}}\otimes\hat{\mathbf{r}}\right)e^{-i\mathbf{k}\cdot\mathbf{r}_i}e^{ik_0 r}, \tag{17}$$

where $\mathbf{k} = k_0\hat{\mathbf{r}}$. $\overleftrightarrow{\mathbf{G}}_{\text{ff}}$ is obtained from Equation (4) keeping only the term of the order of $1/R \approx 1/r$. The expression of $\overleftrightarrow{\mathbf{G}}_{\text{ff}}$ is valid when $R$ is much larger than the wavelength.

## 2.2. Solve the Lippmann–Schwinger Equation and Computation of the Far-Field

We, therefore, have to solve a system of linear equations with $3N$ unknowns (see Equation (15)). However, inverting the dense matrix $(\mathbf{I} - \mathbf{A}\mathbf{D}_\alpha)$ to find $\mathbf{E}_l$, is untenable both in terms of computation time and matrix storage. The solution is therefore to solve this linear system iteratively using a conjugate gradient method. Many iterative methods have been tested and it appears that the most efficient methods are the quasi-minimal residual [38], the general product bi-conjugate Gradient [39] or the bi-conjugate gradient stabilized [40] (one can also use the generalized minimal residual method but it is difficult to manage the memory when $N$ becomes large [41]). On the other hand, iterative methods require one or two matrix-vector products (MVP) to be performed at each iteration. The matrix being dense and of size $3N \times 3N$, the computation time would still be very long. However, Draine and Flatau have proposed to strongly reduce the computation time by using the Toeplitz nature of the matrix (i.e., the elements of the matrix depend only on $(\mathbf{r}_i - \mathbf{r}_j)$) to perform the MVP with a 3D fast Fourier transform (FFT) [42,43].

Once the field inside the object is obtained it is easy to quickly obtain the near field in the vicinity of the object with only one MVP performed with the three-dimensional FFT [37].

Calculating the far-field using Equation (16) for large values of $N$, would require long computation times. Recently, we have proposed a new way to compute the far-field. The Green's function $G_{\mathrm{ff}}$ can be rewritten by transforming the exponential as $e^{-i\mathbf{k}.\mathbf{r}_i} = e^{-i\mathbf{k}_\parallel.\mathbf{r}_{\parallel,i}} e^{-ik_z z_i}$ with $\mathbf{r}_i = (\mathbf{r}_{\parallel,i}, z_i)$, $\mathbf{k} = (\mathbf{k}_\parallel, k_z)$ and $k_z = \sqrt{k_0^2 - \mathbf{k}_\parallel^2}$. Under this new formulation, the summation over the dipoles associated with the location at the layer $z_i$ can then be performed by a two-dimensional FFT. This drastically decreases the computation time by computing the diffracted field in many directions of observation for a large number of subunits [44].

### 2.3. Examples of Calculation with the DDA

In this paragraph, we will show a typical example of the convergence of the DDA. Note that in this article all the results presented are computed with IF-DDA code [10] with by default the polarizability described by Equation (14). The code uses the FFTW (fast Fourier transform in the west) [45] and it is parallelized with OpenMP and runs on 36 processors. For the first study, we have chosen a sphere of radius $r = \lambda$ with a relative permittivity $\varepsilon = 2.5$. We calculate the relative error between the extinction cross section computed with the DDA [6,46] and that calculated with Mie theory versus the number $N$ of subunits used to represent the sphere, see Figure 2a with crosses. We see a rather strong error at the beginning, because of the approximation that the electric field is constant on a subunit, and the approximation to render a spherical shape with a cubic mesh. As $N$ increases the extinction cross section is better computed and the relative error tends to zero. Generally, for the DDA to have a satisfactory accuracy, it is sufficient to have between 5 or 10 meshes per wavelength in the medium considered, typically the mesh size must satisfy the following relationship:

$$k_0 d |n| < 0.5, \tag{18}$$

where $n$ is the refractive index of the object. Note that $d$ should also be smaller than any characteristic size of the object. For metals, due to the high imaginary part of the index, it may be necessary to discretize the object more finely to take into account the skin depth [6,47,48]. With circles, we have plotted the error on the extinction cross section when we use the filtered Green's tensor [7,18,49]. We can see that with the filtered Green's tensor the accuracy of the DDA has been increased.

Almost all the computation time for the DDA is spent in the resolution of Equation (15). This resolution is conducted iteratively and we define the residual of the iterative method as

$$r = \frac{\|\mathbf{E}_{\mathrm{est}} - \mathbf{E}_{\mathrm{ref}} - \mathbf{A}\mathbf{D}_\alpha \mathbf{E}_{\mathrm{est}}\|}{\|\mathbf{E}_{\mathrm{ref}}\|}, \tag{19}$$

where $\mathbf{E}_{\mathrm{est}}$ is the local field estimated with the iterative method. The iterative process is terminated once $r < \eta$, where $\eta$ is a prescribed tolerance. In Figure 2b, we have plotted the number of MVP need to solve iteratively the Lippmann-Schwinger equation as a function of $N$ for $\eta = 10^{-4}$. The important thing is to know that the computation time obviously depends on $N$ and that the bigger $N$ is, the bigger the computation time is. However, the number of MVP needs to solve the linear equation system iteratively depends weakly on $N$ and with the filtered Green's tensor this number is reduced [19], particularly for small $N$.

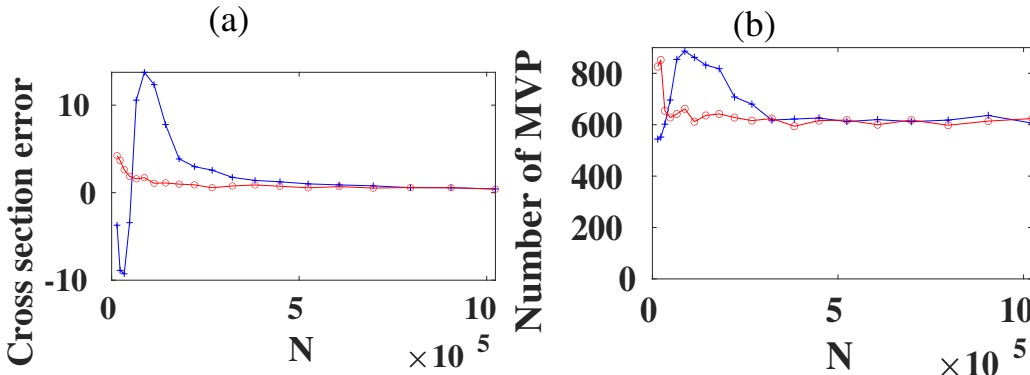

**Figure 2.** We study a sphere of radius $r = \lambda$ and relative permittivity $\varepsilon = 2.5$. With crosses (and blue line) the computation is conducted with the classical DDA, i.e., with Equation (4) for the Green function and (14) for the polarizability, and with circles (and red line) DDA with the Filtered Green's tensor. (**a**) Relative error in per cent on the extinction cross section between the computation conducted with the DDA and the Mie theory versus the number of subunit $N$. (**b**) Number of MVP needs to solve iteratively Equation (15) versus the number of subunit $N$.

To study in more detail the influence of the object's characteristics and of $\eta$ on the number of MVPs, Table 1 presents the number of MVP used by some classical iterative methods for different values of $\eta$ for a sphere of radius $r$ and relative permittivity $\varepsilon$. It is clear that the greater the permittivity and the larger the object, the more difficult it is to achieve convergence. We can see that the best methods are all the Generalized Product (GP) method , the Biconjugate gradient stabilized method (BICGSTAB), and quasi-minimal residual (QMR) method. An extensive study on objects much larger than the wavelength of illumination was carried out in [50] and shows the difficulty of dealing with large objects with high permittivity for the iterative method.

**Table 1.** Number of MVP for different iterative method for two values of $\eta$ to solve Equation (15). The object is a sphere of radius $r$ and relative permittivity $\varepsilon$. The mesh size is $d = 2r/30$ for $r = \lambda$ ($N = 14{,}328$) and $d = 2r/50$ for $r = 2\lambda$ ($N = 65{,}752$). The symbol - means that the convergence of the iterative method has not been achieved. The computation are conducted with the IF-DDA code [10] and the maximum number of MVPs for iterative methods has been set at 10,000.

| Iterative Method | $r = \lambda, \varepsilon = 1.5$ $\eta = 10^{-4}(10^{-8})$ | $r = \lambda, \varepsilon = 3$ $\eta = 10^{-4}(10^{-8})$ | $r = 2\lambda, \varepsilon = 1.5$ $\eta = 10^{-4}(10^{-8})$ | $r = 2\lambda, \varepsilon = 3$ $\eta = 10^{-4}(10^{-8})$ |
|---|---|---|---|---|
| GPBICG1 [51] | 18 (34) | 230 (480) | 42 (74) | 4946 (8712) |
| GPBICG2 [51] | 18 (34) | 238 (474) | 42 (74) | 4676 (8912) |
| GPBICGplus [52] | 18 (34) | 240 (494) | 42 (78) | 4870 (8858) |
| GPBICGsafe [53] | 18 (34) | 240 (476) | 42 (78) | 4902 (8814) |
| GPBICGAR [51] | 18 (34) | 234 (496) | 42 (78) | 5070 (8638) |
| GPBICGAR2 [51] | 18 (34) | 218 (478) | 42 (78) | 5038 (8572) |
| BICGSTAR-plus [54] | 18 (34) | 240 (498) | 42 (78) | 4902 (8804) |
| GPBICOR [55] | 18 (34) | 242 (506) | 44 (74) | 5262 (9116) |
| CORS [56] | 22(38) | - (-) | 44 (82) | - (-) |
| QMR [57,58] | 35 (59) | 329 (519) | 85 (143) | 5313 (7607) |
| TFQMR [58] | 24 (40) | - (-) | 60 (87) | - (-) |
| BICGSTAB [58] | 18 (34) | 258 (538) | 42 (78) | 5316 (9620) |
| QMRBICGSTAB2 [59] | 20 (36) | 378 (-) | 52 (84) | - (-) |
| QMRBICGSTAB1 [59] | - (-) | 670 (696) | - (-) | - (-) |

Note that Table 1 has been calculated with Equation (4) for the Green's function and Equation (14) for the polarizability. Table 2 shows the number of iterations for a sphere of radius $r = 2\lambda$ and $\varepsilon = 3$ with Equations (4) and (14) as a reminder, then the DDA with the Green's function of Equation (4) and the polarizability given by Equation (A5), and finally the DDA with the filtered Green's tensor. With the polarizability given by Equation (14), it can be seen that the number of MVPs to solve Equation (15) for $\eta = 10^{-4}$ is decreased with a better accuracy on the calculation of the extinction cross section. With the filtered Green's tensor the gain in the number of MVPs is large and the accuracy on the extinction cross section is excellent [60].

**Table 2.** Number of MVP for different iterative method for two values of $\eta$ to solve Equation (15) and different version of the DDA. The object is a sphere of radius $r = 2\lambda$ and $\varepsilon = 3$. The symbol - means that the convergence of the iterative method has not been achieved. The computation are conducted with the IF-DDA code [10] and the maximum number of MVPs for iterative methods has been set at 10,000. The last line of the table presents the error in per cent on the extinction cross section, $C_{\text{ext}}$, between the DDA and the Mie theory.

| Iterative Method | DDA with Equations (4) and (14) $\eta = 10^{-4}(10^{-8})$ | DDA with Equations (4) and (A5) $\eta = 10^{-4}(10^{-8})$ | DDA with Filtered Green's Tensor $\eta = 10^{-4}(10^{-8})$ |
|---|---|---|---|
| GPBICG1 [51] | 4946 (8712) | 3412 (9072) | 2284 (6676) |
| GPBICG2 [51] | 4676 (8912) | 3188 (8848) | 2212 (6754) |
| GPBICGplus [52] | 4870 (8858) | 3270 (9348) | 2280 (6560) |
| GPBICGsafe [53] | 4902 (8814) | 3340 (9092) | 2290 (6946) |
| GPBICGAR [51] | 5070 (8638) | 3294 (9154) | 2376 (6164) |
| GPBICGAR2 [51] | 5038 (8572) | 3310 (8774) | 2290 (6490) |
| BICGSTAR-plus [54] | 4902 (8804) | 3342 (9248) | 2324 (6608) |
| GPBICOR [55] | 5262 (9116) | 3550 (9696) | 2322 (6500) |
| CORS [56] | - (-) | - (-) | - (-) |
| QMR [57,58] | 5313 (7607) | 3993 (7521) | 2529 (5429) |
| TFQMR [58] | - (-) | - (-) | - (-) |
| BICGSTAB [58] | 5316 (9620) | 3902 (-) | 2422 (7396) |
| QMRBICGSTAB2 [59] | - (-) | - (-) | - (-) |
| QMRBICGSTAB1 [59] | - (-) | - (-) | - (-) |
| Relative error on $C_{\text{ext}}$ | 17.3% | 2.7% | 0.8% |

The last case studied consists in computing the extinction cross section of a sphere of radius $r = 2\lambda$ versus the permittivity $\varepsilon$. For high permittivities, the extinction cross section computed with Mie theory shows resonances (see Figure 3a) with plain lines. It can be seen that in order for the DDA to find correctly the position of the resonances and their amplitude, it is necessary to strongly discretize the sphere. Indeed, only the $d = \lambda/100$ discretization finds all the resonances correctly (see Figure 3a) with dotted lines. In fact, to have less than 5% error on the resonances, it is necessary to have the highest discretization $d = \lambda/100$ (see Figure 3b). This shows that in the case of resonant objects the criterion $d < \lambda/(10|n|)$ is no longer sufficient. As underlined in [61] the DDA can perfectly represent Mie resonance only if the size $d$ of the subunit is less than twice the width of the resonance, which can requires significant computation time. Indeed, at the Mie resonance, the field inside the object undergoes rapid variations, and it is necessary to take a small mesh size to take them into account.

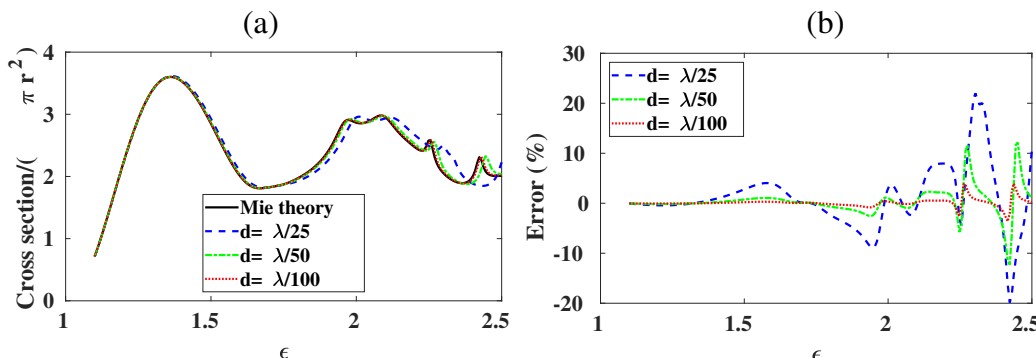

**Figure 3.** We study a sphere of radius $r = 2\lambda$ versus $\varepsilon$. (**a**) Extinction cross section with the Mie theory and DDA for different mesh size $d$. (**b**) Relative error on the extinction cross section between Mie theory and DDA for different mesh size $d$.

Note that in practice we have observed that setting $\eta = 10^{-4}$ is sufficient to accurately obtain the electric field inside the object. In some particular cases, metallic or strongly resonant structures, it may be necessary to take a lower value of $\eta$.

### 3. DDA for an Object in Presence of Multilayer

The DDA historically has been made for objects placed in a homogeneous space, but it is possible to extend it for objects in the presence of a substrate or a multilayer system (see Figure 4). In this case, the most important modification to make is to change the Green's function $\overleftrightarrow{\mathbf{G}}$, by the Green function of the new environment. The computation of the Green's tensor in three-dimensional stratified media composed of an arbitrary number of layers with different permittivities is tricky to perform, but one can have a look at [62] which shows how to efficiently handle the Sommerfeld integrals, by deforming the integration path in the complex plane. This technique makes it possible to take into account waveguides. However, calculating the Green's functions for all pairs $(\mathbf{r}_i, \mathbf{r}_j)$ can be very time consuming for large objects. It is possible in this case to approximate the Green's function using an interpolation of a discrete set of points. Note that only interpolation with rational functions has the ability to accurately take into account the fast decay of the evanescent waves [63]. Note that when the interfaces cross the object then they must pass between the subunits, otherwise the Green's function becomes singular. With a multilayer, in the general case, it is no longer possible to perform the MVP by three-dimensional FFTs. Due to the translational invariance of the multilayer system, the matrix $\mathbf{A}$ is only Toeplitz by block for each pair $(z, z')$. The MVP is then performed by two-dimensional FFTs for each pair $(z, z')$ [44]. However, in the particular case where the object is in the substrate or superstrate, the matrix $\mathbf{A}$ can be written in two parts. The first corresponds to the homogeneous space and the second to the part reflected by the interface. The MVP with the matrix containing the homogeneous space is made with a 3D FFT, while MVP with the matrix containing the interaction with the interface is conducted with 2D FFT for the $x$ and $y$ component and along the $z$-axis we have a discrete correlation, which can be also performed with a 1D FFT [64]. Thus at each iteration, instead of carrying out many two-dimensional FFTs to perform the MVP, it is necessary to do only two three-dimensional FFTs.

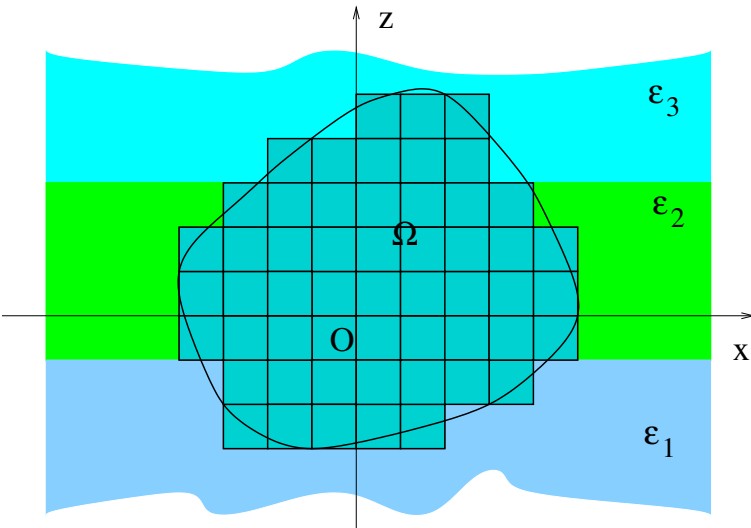

**Figure 4.** Object with a support $\Omega$ inside a multilayer system. When the object contains a interface, it must pass between the discretization elements because Green's function cannot be calculated for a dipole or an observation on the interface.

## 4. Computation of Optical Force

Since the pioneering work on optical force conducted by A. Ashkin [65], in the last decades there has been an increase of interest in the manipulation of small particles by means of the light Lorentz force [66–68]. The time averaged optical force **F** on an object due to the electromagnetic field is computed from Maxwell's stress tensor [1]:

$$\mathbf{F} = 1/(8\pi)\mathrm{Re}\left[\int_S \left[(\mathbf{E}(\mathbf{r}).\mathbf{n})\mathbf{E}^*(\mathbf{r}) + (\mathbf{H}(\mathbf{r}).\mathbf{n})\mathbf{H}^*(\mathbf{r}) - 1/2(|\mathbf{E}(\mathbf{r})|^2 + |\mathbf{H}(\mathbf{r})|^2)\mathbf{n}\right]\mathrm{d}\mathbf{r}\right], \quad (20)$$

where $S$ is a surface enclosing the object, **n** is the local outward unit normal, $*$ denotes the complex conjugate, and Re represents the real part of a complex number. To apply Equation (20) with the DDA, we must first solve the near field equation at each position, and then, compute the electromagnetic field at any position **r** of $S$. This enables us to numerically perform the two dimensional quadrature involved in Equation (20). The problem is, of course, to compute this integration with precision, which requires the calculation of many points on the surface $S$ and therefore a lot of time of computation. We can use at this moment the particularity of the DDA and thus calculate the total force exerted on the object as the sum of the force experienced by each element of discretization. Each subunit acquiring a dipole moment $\mathbf{p}(\mathbf{r}_i)$ under the action of the incident wave, then the $u$-component of the time averaged optical force on the element $i$ can be written as [69]:

$$F_u(\mathbf{r}_i) = \sum_{v=1}^{3} \mathrm{Re}\left(p_v(\mathbf{r}_i)\frac{\partial E_v^*(\mathbf{r}_i)}{\partial u}\right), \quad (21)$$

where $u$ and $v$ stand for the components along either $x$, $y$ or $z$. Then, the net optical force experienced by the object is $\mathbf{F} = \sum_{i=1}^{N} \mathbf{F}(\mathbf{r}_i)$. Notice that to obtain the optical force with Equation (21), it is necessary to know the derivative of the local field at each discretization subunit. On performing the derivative of Equation (14) we obtain:

$$\boldsymbol{\nabla}\mathbf{E}_l(\mathbf{r}_i) = \boldsymbol{\nabla}\mathbf{E}_{\mathrm{ref}}(\mathbf{r}_i) + \sum_{j=1}^{N} \boldsymbol{\nabla}\overleftrightarrow{\mathbf{G}}(\mathbf{r}_i, \mathbf{r}_j)\overleftrightarrow{\alpha}(\mathbf{r}_j)\mathbf{E}_l(\mathbf{r}_j). \quad (22)$$

Notice that the tensor $\boldsymbol{\nabla}\overleftrightarrow{\mathbf{G}}$ has 27 components. The sum over $N$ can be conducted very efficiently with the help of FFT [70]. Many studies on optical forces and optical torques have been carried out with the DDA (see, for example, [14,15,71,72]).

## 5. DDA for Periodic Object

The DDA can also deal with a periodic object, as depicted in Figure 5. The principle is always the same: the periodic object under study is discretized in small subunit and the field at each subunit can be expressed as:

$$\mathbf{E}(\mathbf{r}_i) = \mathbf{E}_{\text{ref}}(\mathbf{r}_i) + \sum_{p=-\infty}^{\infty}\sum_{q=-\infty}^{\infty}\sum_{i=1}^{N}\overleftrightarrow{\mathbf{G}}(\mathbf{r}_i,\mathbf{r}_j + p\mathbf{r} + q\mathbf{v})\overleftrightarrow{\alpha}(\mathbf{r}_j)\mathbf{E}(\mathbf{r}_j + p\mathbf{r} + q\mathbf{v}), \qquad (23)$$

where $\mathbf{u}$ and $\mathbf{v}$ are the lattices vector for the array. Note that the polarizability does not depend on $\mathbf{u}$ and $\mathbf{v}$ because the object is periodic. Solving this system is impossible because of its infinite size, but in the case of plane wave illumination, this equation can be simplified as

$$\mathbf{E}(\mathbf{r}_i) = \mathbf{E}_{\text{ref}}(\mathbf{r}_i) + \sum_{i=1}^{N}\left(\sum_{p=-\infty}^{\infty}\sum_{q=-\infty}^{\infty}\overleftrightarrow{\mathbf{G}}(\mathbf{r}_i,\mathbf{r}_j + p\mathbf{u} + q\mathbf{v})e^{i\mathbf{k}_{\text{ref}}\cdot(p\mathbf{r}+q\mathbf{v})}\right)\overleftrightarrow{\alpha}(\mathbf{r}_j)\mathbf{E}(\mathbf{r}_j), \qquad (24)$$

where $\mathbf{k}_{\text{ref}}$ is the wave vector of the incident wave. Then the self-consistent field on the right-hand side is independent of $(p,q)$ and can be taken out of the infinite sum. Hence, the number of unknowns is now reduced to $N$, the number of subunit of the unit cell, and we need only to change the Green's function. The key point in this calculation is to evaluate the new periodic Green's function. This one being obtained by two infinite sums, it is not possible to realize directly this calculation, the convergence of the series being extremely slow.

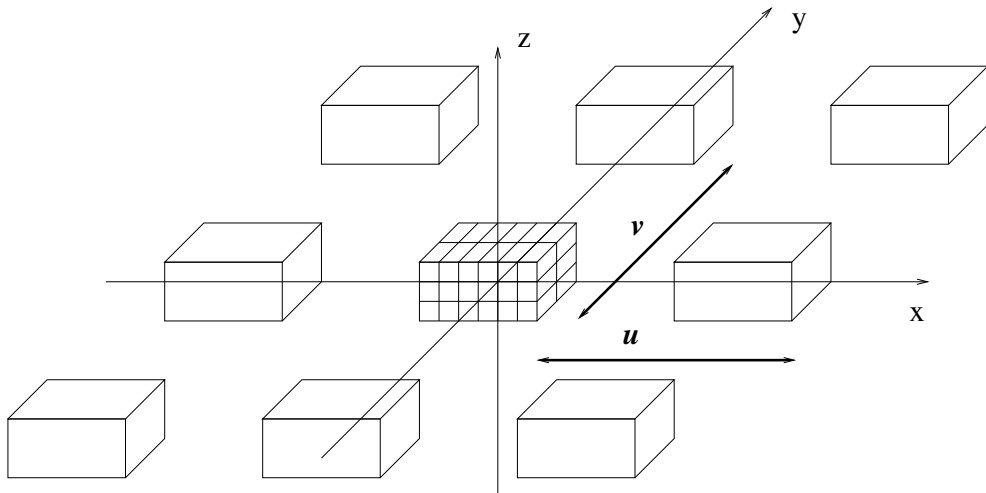

**Figure 5.** Bi-periodic diffraction gratings of period $\mathbf{u}$ and $\mathbf{v}$ in the $(x,y)$ plane. Only the central object is discretized, the other elements are taken into account through the bi-periodic Green's function.

The first solution to compute rigorously the periodic Green's function is to separate the sum into two parts, one conducted in direct space and the other conducted in reciprocal space, so that each sum can be conducted quickly. For more details, the readers can refer to [70,73]. A convergence accelerator can be added to speed up the convergence of the two sums [74].

The second solution to compute the Green's function has been introduced by Draine and Flatau. The double infinite sum is performed in an approximate but much simpler way:

$$\sum_{p=-\infty}^{\infty} \sum_{q=-\infty}^{\infty} \overset{\leftrightarrow}{\mathbf{G}}(\mathbf{r}_i, \mathbf{r}_j + p\mathbf{u} + q\mathbf{v})e^{i\mathbf{k}_{\text{ref}}\cdot(p\mathbf{r}+q\mathbf{v})-(\gamma k_{\text{ref}}r_{j,kpq})^4}, \tag{25}$$

with $r_{j,kpq} = |\mathbf{r}_k + p\mathbf{u} + q\mathbf{v} - \mathbf{r}_j|$. The authors showed that $\gamma = 0.001$ ensures accurate results [75]. We notice that the DDA has also been extended to the simulation of the electromagnetic field scattered by an aperiodic object in presence of a double-periodic structure [76].

## 6. DDA for Magneto-Dielectric Object

The DDA can also deal with scatterers with arbitrary dielectric permittivity and magnetic permeability [77]. The principle remains the same, the object is discretized in small subunit and each subunit is polarized under the action of the incident field and the other subunits, but this time there is an electric dipole moment due to the relative permittivity and a magnetic dipole moment due to the permeability. The electromagnetic field in the object is then written as:

$$\mathbf{E}(\mathbf{r}_i) = \mathbf{E}_{\text{ref}}(\mathbf{r}_i) + \sum_{j=1}^{N} \overset{\leftrightarrow}{\mathbf{G}}^{\text{ee}}(\mathbf{r}_i, \mathbf{r}_j)\overset{\leftrightarrow}{\alpha}^{\text{e}}(\mathbf{r}_j)\mathbf{E}(\mathbf{r}_j) + \sum_{j=1}^{N} \overset{\leftrightarrow}{\mathbf{G}}^{\text{em}}(\mathbf{r}_i, \mathbf{r}_j)\overset{\leftrightarrow}{\alpha}^{\text{m}}(\mathbf{r}_j)\mathbf{H}(\mathbf{r}_j) \tag{26}$$

$$\mathbf{H}(\mathbf{r}_i) = \mathbf{H}_{\text{ref}}(\mathbf{r}_i) + \sum_{j=1}^{N} \overset{\leftrightarrow}{\mathbf{G}}^{\text{me}}(\mathbf{r}_i, \mathbf{r}_j)\overset{\leftrightarrow}{\alpha}^{\text{e}}(\mathbf{r}_j)\mathbf{E}(\mathbf{r}_j) + \sum_{j=1}^{N} \overset{\leftrightarrow}{\mathbf{G}}^{\text{mm}}(\mathbf{r}_i, \mathbf{r}_j)\overset{\leftrightarrow}{\alpha}^{\text{m}}(\mathbf{r}_j)\mathbf{H}(\mathbf{r}_j), \tag{27}$$

$$\text{with } \overset{\leftrightarrow}{\alpha}^{\text{e}} = \overset{\leftrightarrow}{\alpha_0}^{\text{e}}\left(\mathbf{I} - \frac{2}{3}ik^3\overset{\leftrightarrow}{\alpha_0}^{\text{e}}\right)^{-1} \text{and } \overset{\leftrightarrow}{\alpha_0}^{\text{e}} = \frac{3d^3}{4\pi}\left[\overset{\leftrightarrow}{\varepsilon} - \mathbf{I}\right]\left[\overset{\leftrightarrow}{\varepsilon} + 2\mathbf{I}\right]^{-1} \tag{28}$$

$$\overset{\leftrightarrow}{\alpha}^{\text{m}} = \overset{\leftrightarrow}{\alpha}^{\text{m}}\left(\mathbf{I} - \frac{2}{3}ik^3\overset{\leftrightarrow}{\alpha_0}^{\text{m}}\right)^{-1} \text{and } \overset{\leftrightarrow}{\alpha_0}^{\text{m}} = \frac{3d^3}{4\pi}\left[\overset{\leftrightarrow}{\mu} - \mathbf{I}\right]\left[\overset{\leftrightarrow}{\mu} + 2\mathbf{I}\right]^{-1}, \tag{29}$$

where $\overset{\leftrightarrow}{\mathbf{G}}^{\text{ee}}$ is the classical Green's function seen previously in Equation (4) :

- $\overset{\leftrightarrow}{\mathbf{G}}^{\text{ee}}(\mathbf{r}_i, \mathbf{r}_j)\mathbf{p}(\mathbf{r}_j)$ represents the electric field at the position $\mathbf{r}_i$ due to an electric dipole located at $\mathbf{r}_j$.
- $\overset{\leftrightarrow}{\mathbf{G}}^{\text{me}}(\mathbf{r}_i, \mathbf{r}_j)\mathbf{p}(\mathbf{r}_j)$ represents the magnetic field at the position $\mathbf{r}_i$ due to an electric dipole located at $\mathbf{r}_j$.
- $\overset{\leftrightarrow}{\mathbf{G}}^{\text{em}}(\mathbf{r}_i, \mathbf{r}_j)\mathbf{m}(\mathbf{r}_j)$ represents the electric field at the position $\mathbf{r}_i$ due to a magnetic dipole located at $\mathbf{r}_j$.
- $\overset{\leftrightarrow}{\mathbf{G}}^{\text{mm}}(\mathbf{r}_i, \mathbf{r}_j)\mathbf{m}(\mathbf{r}_j)$ represents the magnetic field at the position $\mathbf{r}_i$ due to a magnetic dipole located at $\mathbf{r}_j$.

In Gaussian units we have $\overset{\leftrightarrow}{\mathbf{G}}^{\text{ee}} = \overset{\leftrightarrow}{\mathbf{G}}^{\text{mm}}$ and $\overset{\leftrightarrow}{\mathbf{G}}^{\text{em}} = -\overset{\leftrightarrow}{\mathbf{G}}^{\text{me}}$. The unknowns now consist of $N$ electric fields and $N$ magnetic fields, in taking into account the components, we get $6N$ unknowns. We must therefore solve a linear system of size $6N \times 6N$, which can be written symbolically as:

$$\left[\begin{pmatrix} \mathbf{I} & 0 \\ 0 & \mathbf{I} \end{pmatrix} - \begin{pmatrix} \mathbf{A}^{\text{ee}} & \mathbf{A}^{\text{em}} \\ -\mathbf{A}^{\text{em}} & \mathbf{A}^{\text{ee}} \end{pmatrix}\begin{pmatrix} \mathbf{D}_{\alpha\text{e}} & 0 \\ 0 & \mathbf{D}_{\alpha\text{m}} \end{pmatrix}\right]\begin{pmatrix} \mathbf{E} \\ \mathbf{H} \end{pmatrix} = \begin{pmatrix} \mathbf{E}_{\text{ref}} \\ \mathbf{H}_{\text{ref}} \end{pmatrix}, \tag{30}$$

where $\mathbf{A}$ is the matrix which contains all the dyadic tensors. This system of linear equations is again solved iteratively, and in this case, the best method to use is GPBICG [39]. The DDA with this configuration therefore makes it possible to study objects such as invisibility cloaks, concentrators, etc. [78], which have anisotropic permeability and permeability. Notice that, all the previous advances in DDA presented in this article, can be extended to objects with permittivity and permeability.

### 7. Fluorescence Lifetime Calculations

We want to derive the damping rates of a particle (atom or molecule) in a complex environment. From linear response theory the damping rate of an atom located at $\mathbf{r}_0$ normalized with respect to the free space value can be written as [79–81]:

$$\frac{\Gamma_u}{\Gamma_0} = \frac{3}{2k_0^3}\text{Im}[H_{uu}(\mathbf{r}_0, \mathbf{r}_0)], \tag{31}$$

where $\Gamma_0$ is the free-space decay rate, $H_{uu}$ the diagonal term of the Green's function of the environment, and $u$ stands for $x$, $y$ or $z$. In homogeneous space we get $\overleftrightarrow{\mathbf{H}} = \overleftrightarrow{\mathbf{G}}$, in using the fact that the imaginary part of the Green function of the homogeneous space is perfectly defined at the position of the dipole, i.e., $\text{Im}[\overleftrightarrow{\mathbf{G}}(\mathbf{r}_0, \mathbf{r}_0)] = \frac{2}{3}k_0^3\mathbf{I}$, we obviously get $\frac{\Gamma_u}{\Gamma_0} = 1$. Note that the lifetime is the inverse of the damping rate. While the DDA is usually written to calculate the electromagnetic field, we can also write it to calculate the Green's function of a dipole located at $\mathbf{r}_0$ in presence of an object of arbitrary shape and permittivity [80,82]. The principle is to first calculate the Green's function between the dipole and all the subunit of the object, in presence of the object. These Green's function are solution of the following self-consistent equation:

$$\overleftrightarrow{\mathbf{H}}(\mathbf{r}_i, \mathbf{r}_0) = \overleftrightarrow{\mathbf{G}}(\mathbf{r}_i, \mathbf{r}_0) + \sum_{j=1}^{N}\overleftrightarrow{\mathbf{G}}(\mathbf{r}_i, \mathbf{r}_j)\alpha(\mathbf{r}_j)\overleftrightarrow{\mathbf{H}}(\mathbf{r}_j, \mathbf{r}_0), \tag{32}$$

for $i = 1, \cdots, N$. We have the same system of linear equations as before, except that the incident field is replaced by the free Green's function between the position of the particle and the position of the subunit of the object. Now, if we want to know the Green's function in presence of the object at position $\mathbf{r}_0$ for a dipole at $\mathbf{r}_0$, we get:

$$\overleftrightarrow{\mathbf{H}}(\mathbf{r}_0, \mathbf{r}_0) = \overleftrightarrow{\mathbf{G}}(\mathbf{r}_0, \mathbf{r}_0) + \sum_{j=1}^{N}\overleftrightarrow{\mathbf{G}}(\mathbf{r}_0, \mathbf{r}_j)\alpha(\mathbf{r}_j)\mathbf{H}(\mathbf{r}_i, \mathbf{r}_0). \tag{33}$$

If the object is in the presence of a substrate or a multilayer, $\overleftrightarrow{\mathbf{G}}$ must be replaced by the Green's function of the reference system. Notice the decay rate is also strongly related to $\rho(\mathbf{r}_0)$, the local density of state (LDOS) [81,83]:

$$\rho(\mathbf{r}_0) = \frac{3}{2k_0\pi^2 c}\text{Im}\left\{\text{Tr}[\overleftrightarrow{\mathbf{H}}(\mathbf{r}_0, \mathbf{r}_0)]\right\}, \tag{34}$$

where Tr denotes the trace of the matrix. Notice that the LDOS in vacuum reduces to $\rho(\mathbf{r}_0) = k_0^2/(\pi^2 c)$ [81]. Thus, the DDA can map the LDOS in complex nanostructures.

### 8. DDA in Time Domain

In the time domain, electromagnetic scattering is usually addressed using the finite difference in time domain (FDTD) method [84]. In order to have the temporal evolution of the electromagnetic field, the FDTD principle consists in discretizing the whole space of interest into a set of elementary cells. The values of the electric and magnetic fields are calculated at each point of the structure and at each instant. The time steps lead to small numerical errors, but at long times the numerical dispersion can severely degrade the accuracy and even be totally unacceptable [85]. The DDA has been extended to the temporal regime. It means that material anisotropy and dispersion are easily taken into account. The main advantage of the DDA in temporal regime is that, unlike the FDTD, the global error, on the computed fields, depends mainly on the spatial discretization of the object, since the space between scatterers need not be discretized. The principle is as follows: the magnitude of the incident field which illuminates the object is written as $\mathcal{F}(t)$ and has the spectrum $F(\omega)$ which then corresponds to the Fourier transform of $\mathcal{F}(t)$. The

Fourier transform of the incident field $\mathcal{E}_{\text{ref}}(\mathbf{r}, t)$ at any point in space, i.e., $F(\omega)\mathbf{E}_{\text{ref}}(\mathbf{r}, \omega)$, allows us to obtain the incident field in harmonics. It is then sufficient to use the DDA to know the field in all space at many frequencies and by a simple inverse Fourier transform one obtains the field in time domain at any point [86]. In practice, $M$ values are necessary in the frequency domain for $F(\omega)$ in accordance with the Nyquist-Shannon sampling theorem. Then, the DDA has to be applied $M$ times, which may seem a bit tedious, but it is possible when calculating the frequency $n$ to use the result of the previous $n-1$ frequency or several previous frequencies to get a good initial guess for the iterative method and thus save a lot of calculation time. Notice that an other approach has been conducted by Kim and Yurkin [87]. They develop a time domain DDA, describing the temporal evolution of electric field for plasmonic nanoparticles.

## 9. Conclusions

The great strength of the DDA is that it can handle objects with arbitrary shapes and permittivities, which can also be anisotropic. With the DDA it is necessary to discretize only the object and the outgoing wave conditions are automatically satisfied through the Green's function. It is therefore not necessary to use perfectly matched layers as with finite difference or finite elements. Moreover, the method is very versatile as it can easily adapt to a wide range of different configurations: object in the presence of a multilayer, object with a permittivity and permeability, in the time domain, study of the lifetime of fluorescent molecules, periodic objects, etc. This simple method in its initial conception has evolved over time into a multifunctional toolkit and there is now a lot of DDA-based code available on the net. In Appendix B we give a list (not exhaustive), of freely available codes and their features.

Of course, DDA has its drawbacks and limitations. DDA requires solving a large system of linear equations with a dense matrix. Even with an iterative method, and with the help of FFTs to perform the MVP, it is still complicated to handle objects of several tens of wavelengths, especially if the permittivity is high. This was shown in [50] where Yurkin *et al.* plot the convergence as a function of the relative permittivity and object size and clearly show that the larger the object, the slower the convergence of the iterative method. We also note that for a high permittivity ($|\varepsilon| > 10$) that the field inside the object exhibits oscillations that cause the accuracy of the DDA to drop [88–90]. We also note that for metals then the accuracy of the method starts to drop and for perfect conductors the DDA can no longer be used. This is because the DDA calculates the field inside the object and in the case of perfect conductors this is zero! In this case, surface methods should be preferred.

In the future, the DDA should be improved, in particular, it will therefore be necessary to focus on its shortcomings, particularly on the resolution of the system of linear equations, for example with a preconditioner to accelerate the iterative method [91,92] or by improving the initial estimate of the iterative method [93,94]. This would make it possible to deal with larger objects. The problem of convergence of the DDA for objects with high permittivities will also have to be studied. A solution has been proposed for simple objects as the tracks currently proposed only apply to spheres or ellipsoids [30,95] (this is achieved by changing the polarizability of the dipoles on the surface of the object that accounts for local-field effects) but remains to be conducted for objects of arbitrary shapes.

In general, when choosing a numerical method to calculate electromagnetic diffraction rigorously, it is necessary to define the configuration, the object under study, in order to choose the most appropriate method. It is also possible to use approximate numerical methods in electromagnetism, such as the Born approximation, the Rytov approximation [96], beam propagation method [97,98], etc. These methods are simple to implement, but have very precise conditions of application, as determined by the approximations which were used to establish them [99].

**Funding:** This research received no external funding.

**Institutional Review Board Statement:** Not applicable.

**Informed Consent Statement:** Not applicable.

**Data Availability Statement:** All the data presented in this article are obtained form IF-DDA codes and can therefore be recomputed with this code freely available on the net. See Appendix B for the internet address.

**Acknowledgments:** I thank B. Stout for a careful reading of this manuscript.

**Conflicts of Interest:** The author declares no conflict of interest.

## Appendix A. Integration of the Singularity of the Green's Function

With the DDA we should perform the following integration which represents the interaction of the mesh on itself, *i. e. j = i* in Equation (7) :

$$\mathbf{K}_{V_0} = \int_{V_0} \overleftrightarrow{\mathbf{G}}(\mathbf{r}_0, \mathbf{r}') \overleftrightarrow{\chi}(\mathbf{r}') \mathbf{E}(\mathbf{r}') \mathrm{d}\mathbf{r}', \tag{A1}$$

with $\mathbf{r}_0 \in V_0$. For example, with DDA the volume $V_0$ is often cubic with $\mathbf{r}_0$ at the center of the cube. Obviously when $\mathbf{r}' = \mathbf{r}_0$, the Green's function $\overleftrightarrow{\mathbf{G}}(\mathbf{r}_0, \mathbf{r}_0)$ is undefined because it depends on $1/|\mathbf{r}' - \mathbf{r}_0|^3$. Nevertheless, even if the Green's function is singular, the integration represented by Equation (A1) can be computed. The principle is as follow [19,24,100]. Equation (A1) can be rewritten as [24]

$$\begin{aligned} \mathbf{K}_{V_0} &= \int_{V_0} \left[ \overleftrightarrow{\mathbf{G}}(\mathbf{r}_0, \mathbf{r}') \overleftrightarrow{\chi}(\mathbf{r}') \mathbf{E}(\mathbf{r}') - \overleftrightarrow{\mathbf{G}}_s(\mathbf{r}_0, \mathbf{r}') \overleftrightarrow{\chi}(\mathbf{r}) \mathbf{E}(\mathbf{r}) \right] \mathrm{d}\mathbf{r}' \\ &- \int_{S_0} \mathbf{n} \frac{\mathbf{r} - \mathbf{r}_0}{|\mathbf{r} - \mathbf{r}_0|^3} \mathrm{d}^2 \mathbf{r} \overleftrightarrow{\chi}(\mathbf{r}_0) \mathbf{E}(\mathbf{r}_0), \end{aligned} \tag{A2}$$

where $\overleftrightarrow{\mathbf{G}}_s = \frac{1}{R^3} \left( 3\hat{\mathbf{R}} \otimes \hat{\mathbf{R}} - \overleftrightarrow{\mathbf{I}} \right)$ is the static Green function, i.e., $\overleftrightarrow{\mathbf{G}}$ with $k_0 = 0$, $S_0$ the surface of $V_0$ and $\mathbf{n}$ the external normal to the surface $S_0$. The integration of $\overleftrightarrow{\mathbf{L}} = \int_{S_0} \mathbf{n} \frac{\mathbf{r} - \mathbf{r}_0}{|\mathbf{r} - \mathbf{r}_0|^3} \mathrm{d}^2 \mathbf{r}$ gives a real symmetric dyadic tensor with a trace equal to $4\pi$. In the case of a cubic or spherical shape we get $\overleftrightarrow{\mathbf{L}} = \frac{4\pi}{3} \overleftrightarrow{\mathbf{I}}$ [22], when the shape is a cuboid the expression is more complex [22]. Assuming $\overleftrightarrow{\chi}(\mathbf{r}') \mathbf{E}(\mathbf{r}')$ constant over the subunit we get:

$$\mathbf{K}_{V_0} = \left[ \overleftrightarrow{\mathbf{M}} - \overleftrightarrow{\mathbf{L}} \right] \overleftrightarrow{\chi}(\mathbf{r}_0) \mathbf{E}(\mathbf{r}_0) \tag{A3}$$

$$\text{with } \overleftrightarrow{\mathbf{M}} = \int_{V_0} \left[ \overleftrightarrow{\mathbf{G}}(\mathbf{r}_0, \mathbf{r}') - \overleftrightarrow{\mathbf{G}}_s(\mathbf{r}_0, \mathbf{r}') \right] \mathrm{d}\mathbf{r}'. \tag{A4}$$

It is complicated to calculate rigorously $\overleftrightarrow{\mathbf{M}}$ for a cube, but an accurate approximation is available in Ref. [23]. However, it is easy to calculate it for a sphere of volume equivalent to a cube of side $d$. In this case we obtain:

$$\overleftrightarrow{\mathbf{M}} = \frac{8\pi}{3} \left[ (1 - ik_0 a) e^{ik_0 a} - 1 \right] \mathbf{I}, \tag{A5}$$

with $\frac{4\pi}{3} a^3 = d^3$. If we write the DDA in term of local field the polarizability of the dipoles can be expressed as [19]:

$$\overleftrightarrow{\alpha} = \overleftrightarrow{\alpha_0} \left( \overleftrightarrow{\mathbf{I}} - \frac{\overleftrightarrow{\mathbf{M}} \overleftrightarrow{\alpha_0}}{d^3} \right)^{-1}. \tag{A6}$$

If we assume that $d$ is equal to zero, we obtain $\overleftrightarrow{\mathbf{M}} = \overleftrightarrow{\mathbf{0}}$ and the Clausius–Mossotti polarizability for the dipoles. If we assume $d$ to be small ($d \ll \lambda$) and perform a Taylor series of $\overleftrightarrow{\mathbf{M}}$ and keep only the imaginary part, we obtain $\overleftrightarrow{\mathbf{M}} = \frac{2}{3} i k_0^3 d^3$ and Equation (14) for the polarizability.

## Appendix B. Open-Source DDA Codes

There are many DDA-based codes available on the network. Each code has its own particularities and allows the study of different physical data in different environments. Table A1 summarises the possibilities of the best-known codes.

**Table A1.** This table presents the most well-known DDA codes with the different calculations to which they give access. The last line of the table presents which language is used for parallelization.

| Code | DDscat | ADDA | IF-DDA(M) | openDDA | MPDDA | DDA-SI |
|---|---|---|---|---|---|---|
| Periodic target | × | | | | | |
| Optical force | × | × | × | | | × |
| Cross section | × | × | × | × | × | × |
| Substrate | | × | × | | | × |
| Multilayer | | | × | | | |
| Microscopy | | | × | | | |
| LDOS | | × | | | | |
| Muller matrix | × | × | | | | |
| Anisotropy | × | × | × | | | × |
| Parallel | openMP MPI | MPI | OpenMP | openMP MPI | Matlab | Matlab |

Below we list the different codes, with the names of the authors, and a short note on the code.

- DDscat: Draine and Flatau [17]. DDscat is the best known and oldest code. The current version is DDSCAT 7.3.3 and the code is written in FORTRAN 90. You can download the code on the following page: http://ddscat.wikidot.com/, accessed on 12 July 2022.
- ADDA : Yurkin et al. [101]. ADDA is a a C software package and can also employ GPUs to accelerate computations. You can download the code on the following page: https://github.com/adda-team/adda, accessed on 12 July 2022.
- IF-DDA(M) : Chaumet, Sentenac and Sentenac [10]. The code actually consists of two. One where the object is in a homogeneous space (IF-DDA) and another one that allows to put the objects in a multilayer (IF-DDAM). The codes are in FORTRAN and the interface with drop-down menus that makes it suitable for non-physicists is coded in C++. You can download the code on the following page: https://www.fresnel.fr/perso/chaumet/ifdda.html or https://gitlab.com/ifdda, accessed on 12 July 2022.
- openDDA: Mc Donald, Golden, and Jennings [102]. This code is highly optimized in terms of celerity and memory. You can download the code on the following page: https://github.com/drjmcdonald/OpenDDA, accessed on 12 July 2022.
- MPDDA : Shabaninezhada, Awan and Ramakrishnac [103]. This package developed in MATLAB has been conducted to reduce the computational cost and can use the parallel computing toolbox in MATLAB by running the codes in GPU. You can download the code on the following page: https://github.com/MasoudShabani/MPDDA-1.0, accessed on 12 July 2022.
- DDA-SI : Loke, Mengüç and Nieminen [104]. DDA with surface interaction for MATLAB. You can download the code on the following page: https://github.com/dalerxli/dda-si, accessed on 12 July 2022.

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
