# Peer review of "The Discrete Dipole Approximation: A Review"

_mathematics, doi:10.3390/math10173049_

Round 1
Reviewer 1 Report
The manuscript presents a relatively concise review of the DDA covering its basics and several extensions. It is easy to read and is definitely suitable for the special issue in Mathematics. I have a number of comments aimed at further improvement of the review:
1) Please specify the used system of units (seems to be Gaussian), for instance, in the end of Introduction.
2) I recommend to mention other reviews on the same subject in the Introduction. This may include Refs. 70, 25 and the following chapter:
M. A. Yurkin, “Computational approaches for plasmonics,” in Handbook of Molecular Plasmonics, F. Della Sala and S. D’Agostino, Eds., pp. 83–135, Pan Stanford Publishing, Singapore (2013).
3) Eq.(5) appears out of nowhere, and it is important to note that it has a strongly singular kernel. At least, the author should mention that certain exclusion volume is used (together with some singular L-term). It is briefly discussed later, but more details are needed. Ref.15 is appropriate here, but the author may consider also a more recent:
M. A. Yurkin and M. I. Mishchenko, “Volume integral equation for electromagnetic scattering: Rigorous derivation and analysis for a set of multilayered particles with piecewise-smooth boundaries in a passive host medium,” Phys. Rev. A 97(4), 043824 (2018) [doi:10.1103/physreva.97.043824].
4) lines 59-60: the dipole should also be smaller than any characteristic size of the object. The same applies to line 131 and Eq.(18).
5) With regards to Fig.2, the author should provide more details on simulation parameters. Which code and what DDA (polarizability) formulation were used? How the parallelization over 36 processors (line 137) was performed? For instance, I ran a test simulation with the sequential version of ADDA on a single core of my laptop:
adda -lambda 1 -size 2 -m 1.5811 0 -eps 4 -grid 100 -pol cm
which has 5.2e5 dipoles in the sphere (corresponding to the middle of Fig.2b) and took 77 seconds. This suggests that data on Fig.2 corresponds to sequential rather than parallel execution.
6) Related to the previous one. The author should double check the results corresponding to peaks in Fig.2b. First, the explanation with regards to prime decomposition of grid size is not convincing, since the FFTW is known to handle even large prime numbers reasonable well. Second, I have tried to reproduce the results with ADDA (which also uses FFTW):
adda -lambda 1 -size 2 -m 1.5811 0 -eps 4 -grid 67 -pol cm
adda -lambda 1 -size 2 -m 1.5811 0 -eps 4 -grid 70 -pol cm
this corresponds to 1.6e5 and 1.8e5 dipoles in the sphere (around the largest peak in Fig.2b) and computational time 21.9 and 22.8 seconds, respectively. So using prime number 67 for the grid size does not seem to cause any problems in the FFTW. I suggest the author to look at the number of iterations in his data. Normally, Niter should be almost independent of N, but unusually bad cases may potentially appear for specific combinations of problem parameters.
7) Similarly to the above, the results of Table 1 should specify the used DDA formulation (polarizability and interaction term, is it the simplest Clausius-Mossotti one – Eq. (12)?). I have approximately reproduced the largest number of iterations with ADDA (4065 iterations):
adda -lambda 1 -size 4 -m 1.732 0 -eps 4 -grid 50 -pol cm
I have also tried IGT and FCD formulations. While the IGT led to only marginal improvement (3854 iterations), the FCD helped a lot (1139 iterations):
adda -lambda 1 -size 4 -m 1.732 0 -eps 4 -grid 50 -pol fcd -int fcd
So the author should at least mention that using modern formulations for the interaction term may partly alleviate this issue. This has already been mentioned in (last paragraph of Section 3 of)
C. Liu et al., “Comparison between the pseudo-spectral time domain method and the discrete dipole approximation for light scattering simulations,” Opt. Express 20(15), 16763–16776 (2012) [doi:10.1364/OE.20.016763].
8) The author may also consider mentioning that certain (analytical) estimates of the number of iterations are available. See the chapter (Yurkin 2013) mentioned above, as well as
M. A. Yurkin, “Performance of iterative solvers in the discrete dipole approximation,” in 2016 URSI International Symposium on Electromagnetic Theory (EMTS), pp. 488–491, IEEE Press, Espoo, Finland (2016) [doi:10.1109/URSI-EMTS.2016.7571433].
9) Related to the previous ones, the DDA formulations may also greatly affect the accuracy, which is worth mentioning. Consider, e.g., the rightmost peak on Fig.3b (refractive index m=1.557, Lorenz-Mie value for extinction efficiency Qext=2.312). I have run the following tests with ADDA:
adda -lambda 1 -size 4 -m 1.557 0 -eps 4 -grid 100 -pol cm
adda -lambda 1 -size 4 -m 1.557 0 -eps 4 -grid 100 -pol igt_so -int igt 5
adda -lambda 1 -size 4 -m 1.557 0 -eps 4 -grid 100 -pol fcd -int fcd
The corresponding results are for Qext are 2.010 (agrees with blue line in Fig.3a), 2.242, and 2.318, respectively. The accuracy of the FCD is really surprising. So the author should at least mention such possibility.
Also, since the author discusses the feasibility of reproducing the Mie resonances with the DDA, the following paper seems largely relevant:
Y. Zhu, C. Liu, and M. A. Yurkin, “Reproducing the morphology-dependent resonances of spheres with the discrete dipole approximation,” Opt. Express 27(16), 22827–22845 (2019) [doi:10.1364/OE.27.022827].
10) Eq.(31) appears out of nowhere. Please provide some reference
11) In the conclusion the author discusses potential way to accelerate iterative solver and various approximations for the internal field. Then it is also worth to mention a recent hybrid option, which uses the WKB approximation for internal field as an initial guess of the iterative solver:
K. G. Inzhevatkin and M. A. Yurkin, “Uniform-over-size approximation of the internal fields for scatterers with low refractive-index contrast,” J. Quant. Spectrosc. Radiat. Transfer 277, 107965 (2022) [doi:10.1016/j.jqsrt.2021.107965].
There are also several minor issues:
a) line 11: “at several hundred wavelengths” – not clear, please rephrase.
b) line 39: “verifies” -> “satisfies”
c) line 85: “[17]” -> Ref. 16 is also appropriate here (as it contains accurate approximation of the self-term).
d) line 132: “meshsize” -> “mesh size”
e) Tables 1 and 2 does not fit on the page, at least when printed on paper.
f) line 156: “r=λ” -> “r=2λ”.
g) line 172: “…change is to change” – please rephrase.
h) line 330: “open-source” -> “Open-source”
i) line 340: “Yurkin” -> “Yurkin and others”, since there are at least 10 contributors mentioned at https://github.com/adda-team/adda/graphs/contributors, and many more – at https://github.com/adda-team/adda/wiki/Acknowledgements#contributors .
Reviewer 2 Report
The author in this work presents a review of the discrete dipole approximation (DDA) method of calculating electromagnetic diffraction. The manuscript is mostly well written, however, it does have some issues:
1) Similarity index is very high (36%). This needs to be drastically reduced.
2) Self-citation ratio is very high (~27%), especially for a review paper. This also needs to be reduced.
3) Do not use “Refs”, just brackets for citations throughout the manuscript. E.g. correct line 20 “…to Refs [3] and [4]” to “ .. to [3] and [4]”.
4) All figures and tables should be cited in the main text as Figure 1, Table 1, etc. E.g line 53 Fig.1(a).
5) Lines 292-307 are mainly copied from [68].
6) In Equation (20) periods are used.
7) Line 320 “…periodic objects,…” . Use “etc.” instead of periods.
8) Line 327 “conductorss” correct double s
9) Line 330 Capitalize 1st letter in open-source
10) Explain all acronyms at first use (e.g PML ).
11) In the strengths and weaknesses section, which plays the role of the discussion section, there are claims without backing data (e.g. for large permittivity the accuracy starts to drop). This is not to say that the statements are wrong, but rather that statements in a review paper should be backed with data or figures or at least a specific reference (ideally multiple).
12) Section 10 does not really contribute to the manuscript since it does not compare the codes and serves more as a means to present the download links. In its present form, it could be added as an appendix.
13) Conclusion section should be rewritten. I suggest adding its current text to the strength and weaknesses section (along with a more detailed presentation of the problems).
14) Format tables correctly (According to the template)
15)The Abstract needs some rework too. Especially the sentence regarding the developments over the last 20 years is misleading. Almost 50% of citations are before 2000; moreover, nowhere in the paper is the discussion made in the light of structured method developments but solely from various applications' points of view. See also the earlier comment on the codes.
Round 2
Reviewer 1 Report
The authors have addressed all my comments, and the manuscript has become even better. I have only a few minor remarks about the added material.
a) Lines 134-135: FFTW is the abbreviation for “the Fastest Fourier transform in the West”, but I think that using just FFTW is more conventional.
b) Fig.2: please specify the meaning of two curves (I guess RR and FCD) on the legend (preferable to add one) or in the caption.
c) Line 187: “can requires an important time of computation” -> “can require significant computation time”.
d) Line 385: “B. stout” -> “B. Stout”.
e) Line 399: “It is complicated to calculate rigorously M for a cube”. While that is technically true, an accurate approximation is available in Ref.20. And that is probably accurate enough for all purposes. So, I propose to mention it.
Reviewer 2 Report
The author has addressed my comments with the exception of Comment 4.
Figure callouts should be in the form “Figure X” instead of “Fig. X” according to instructions to the authors and the template given. “All figures and tables should be cited in the main text as Figure 1, Table 1, etc.”
I recommend accepting the paper for publication after the correction of the aforementioned issue.
